# Age-Stage, Two-Sex Life Table of *Leptinotarsa decemlineata* (Coleoptera: Chrysomelidae) Experiencing Cadmium Stress

**DOI:** 10.3390/insects16010073

**Published:** 2025-01-13

**Authors:** Bingyu He, Jiebo Zhang, Yang Hu, Yi Zhang, Jianan Wang, Chao Li

**Affiliations:** Key Laboratory of Prevention and Control of Invasive Alien Species in Agriculture & Forestry of the North-Western Desert Oasis, Ministry of Agriculture and Rural Affairs, College of Agronomy, Xinjiang Agricultural University, Urumqi 830052, China; 18040907242@163.com (B.H.); zhangjiebo9729@163.com (J.Z.); m17716991015@163.com (Y.H.); 15071785641@vip.163.com (Y.Z.); 18322007712@163.com (J.W.)

**Keywords:** cadmium, life table, potato, *Leptinotarsa decemlineata*

## Abstract

The Colorado potato beetle (*Leptinotarsa decemlineata*) (Say), recognized as one of the most destructive quarantine pests globally, poses a severe threat to multiple solanaceous crops. While cadmium has been demonstrated to affect insect growth and development, its specific impact on the growth, development, and population parameters of *L. decemlineata* remains underexplored. This study employed an age-stage, two-sex life table to assess the effects of cadmium exposure on the biological characteristics, population dynamics, and reproductive performance throughout the entire development process of *L. decemlineata*. The results indicated that cadmium stress significantly reduced beetle lifespan and fecundity, prolonged certain developmental stages, and increased the incidence of deformities in newly eclosed adults. These findings suggest that cadmium exposure can negatively affect the growth, development, and population dynamics of *L. decemlineata*, potentially inhibiting population growth.

## 1. Introduction

The Colorado potato beetle (*Leptinotarsa decemlineata*) (Say), a herbivorous insect from the order Coleoptera and family Chrysomelidae, is a highly destructive quarantine pest with global significance. Both adults and larvae can feed on potato leaves and eat the whole leaves. The economic loss of potato species and commercial potato-producing areas can be as high as 50%, and in worst-case scenarios, there will be no harvest whatsoever. The Colorado potato beetle is also a major agricultural quarantine pest in China [1,2]. Originally found in the Rocky Mountains of North America, by 2022, this pest covered an area of 90,000 mu in China, and it is now widely distributed in more than 40 countries in Europe, America, and Asia. Due to its remarkable adaptability to diverse environmental conditions, this pest has caused substantial damage and posed severe threats to China’s potato industry since its introduction through Xinjiang [3]. Cadmium is one of the most toxic heavy metal pollutants, exhibiting strong carcinogenic, teratogenic, and mutagenic properties. It is characterized by high environmental mobility, chemical reactivity, and persistent toxicity, posing serious risks to human health through bioaccumulation in the food chain [4,5]. Due to the rapid pace of industrialization in China, factors such as the extensive use of fertilizers, pesticides, and sludge, industrial wastewater discharge, and increased atmospheric deposition of heavy metals have significantly elevated cadmium concentrations in agricultural soils. Currently, approximately 200,000 km^2^ of soil in China—accounting for one-sixth of the country’s total cultivated land—is contaminated with cadmium [6,7]. The bioaccumulation of cadmium through the food chain has critically compromised the growth and development of both crops and insects and adversely affected crop yields [8,9,10]. Among the heavy metal pollutants found in field crops, cadmium is particularly concerning. Cadmium contamination in soils has degraded the quality of agricultural products, posed significant risks to human health, and undermined agricultural sustainability [11]. In western China, a key potato production region, soil cadmium levels consistently exceed permissible limits, resulting in severe pollution [12].

Soil serves as a crucial interface within agroecosystems, connecting organic and inorganic components. Crops absorb heavy metals from contaminated soils and transfer these pollutants into the food chain, endangering the health of insects, animals, and humans [13]. Plants have the ability to absorb, transport, and accumulate heavy metals such as cadmium [14]. Cadmium exposure disrupts various physiological processes in plants, including seed germination, seedling growth, photosynthesis, and antioxidative functions [15], leading to phytotoxicity. This is evident through symptoms such as leaf curling, chlorosis, and inhibited root and stem development [16,17]. Cadmium stress has been shown to impair photosynthetic activity, notably by reducing chlorophyll (Chl) content, thus suppressing photosynthesis and stunting plant growth [18,19,20] and subsequently affecting the crop productivity [8]. Herbivorous insects play a pivotal role in agricultural ecosystems and biodiversity. They are integral to the transfer of cadmium through the food chain. Upon consuming cadmium-contaminated plants, they ingest cadmium, which may undergo biomagnification [21]. The concentration of cadmium stress and the structure of the food chain are key factors influencing the patterns of cadmium transfer within the ecosystem [22].

Research indicates that cadmium can impact the growth and development of insects, leading to alterations in developmental duration, pupation rate, eclosion rate, pupal weight, and lifespan [23]. The toxicological effects of cadmium stress vary across insect species. For instance, in response to cadmium exposure, the gypsy moth (*Lymantria dispar*) demonstrates prolonged developmental periods, along with reduced larval body length and pupal weight [24]. Houseflies (*Musca domestica*) exhibit decreased hatchability, body weight, and survival rates [25]. In *Drosophila melanogaster*, cadmium exposure shortens the lifespan and reduces reproductive capacity, and these effects are positively correlated with cadmium concentration [26]. Regarding *Sitobion avenae*, combined exposure to cadmium and zinc increases the net reproductive rate in generations 1, 3, 5, and 10, although no significant impact on the average generation time is observed [27]. Heavy metal exposure has also been associated with increased rates of insect deformities [28]. Long-term exposure to heavy metal stress can drive adaptive evolution in insects, enhancing their resistance to environmental stressors and insecticides [29,30]. This heightened resistance can lead to pest population outbreaks, which severely threaten the stability of agricultural and forestry ecosystems and compromise national food security [31,32]. Despite these insights, limited research has been conducted on the cadmium–crop–herbivorous insect food chain and its effects on the population dynamics of herbivorous insects. Understanding the ways in which cadmium contaminates its environment, the adverse effects of *L. decemlineata* on potato plants’ growth and development, and the population parameters of *L. decemlineata* is critical for implementing targeted pest management strategies.

## 2. Materials and Methods

### 2.1. Rearing of Insects and Plants

In May 2024, overwintering adult Colorado potato beetles were collected from potato fields in Louzhuangzi Village, Jimusaer County, Changji Hui Autonomous Prefecture, Xinjiang Uygur Autonomous Region, China (89.15° E, 43.77° N). The beetles were placed in Petri dishes (diameter: 9 cm) and transported to the laboratory. Leaves from pre-cultivated potato plants (variety: ‘Holland 15’) were provided and acted as the host plant. The leaves were replaced regularly. The Petri dishes were cleaned and then transferred to a climate-controlled chamber (manufacturer: Ningbo Jiangnan Instrument Factory, China; model: RXM-168C-1) set to 27 ± 1 °C, with relative humidity of 70% ± 5% and a photoperiod of 16 h of light and 8 h of darkness. Under these controlled conditions, the beetles were reared until the emergence of second-generation adults. Eggs laid by these second-generation adults were collected, and newly hatched first-instar larvae from the same day were selected for subsequent experimental procedures.

### 2.2. Effect of Cadmium on Leptinotarsa decemlineata

The potato cultivation experiment was conducted outdoors in flowerpots (30 cm diameter) using the potato variety ‘Holland 15’. Five cadmium treatments were established, with the following concentrations: CK (0 mg/kg), B30 (30 mg/kg), C60 (60 mg/kg), D90 (90 mg/kg), and E120 (120 mg/kg). A standard solution of the single element cadmium (1000 µg/mL, 50 mL) was obtained from Tongluoma Technology (Beijing, China) Co., Ltd. Each pot was filled with 3 kg of nutrient soil, into which the respective concentration of cadmium solution was thoroughly mixed. The soil was turned daily for 15 days, followed by a 15-day aging period. Then, a single potato seed was sown in each pot. The experiment included five treatments, with 12 replicates per treatment. Post-germination, only one potato plant was retained per pot.

In order to assess the impact of cadmium on *L. decemlineata*, the beetles were reared on potato leaves grown under the five cadmium treatments. For each treatment, 30 first-instar larvae, hatched on the same day and exhibiting uniform vitality, were selected per replicate, with four replicates per treatment. Potato leaves from the corresponding cadmium-treated plants were placed in Petri dishes as food, with their bases wrapped in cotton soaked with distilled water to maintain leaf freshness. Larvae were group-reared in the Petri dishes, feeding on potato leaves which were replaced daily. Their developmental stage and survival were observed and recorded every day. Once the fourth-instar larvae stopped feeding, they were transferred to rearing cups (9 cm diameter and 7.3 cm height) containing 5 cm of moist sand, which was sprayed with distilled water to facilitate pupation and eclosion. Upon adult emergence, males and females that emerged on the same day were paired in a 1:1 ratio. Priority was given to pairing individuals that emerged concurrently. If males were unavailable, substitutes from the same treatment were used. In the event of male mortality, replacements were made from the same treatment, and their survival days were no longer recorded. Similarly, if females died, their egg-laying activity and survival days were no longer tracked. The growth and development of *L. decemlineata* were monitored daily. Larval instars were identified according to the *Guidelines for L*. *decemlineata Monitoring* [33]. The egg production and survival status for each beetle were observed and recorded every day.

### 2.3. Statistical Analysis

Data organization and preliminary statistics were performed using Microsoft Excel, and statistical analysis was conducted using IBM SPSS Statistics 26.0. A one-way analysis of variance (ANOVA) was employed to evaluate differences in potato plant emergence time, plant height, stem diameter, chlorophyll content, yield, and the number of deformities observed in *L. decemlineata*. Duncan’s multiple range test was performed to identify differences within groups. Key developmental metrics of *L. decemlineata*, including the developmental period, survival rate, population parameters (innate growth rate *r*, finite growth rate *λ*, net reproductive rate *R*_0_, mean generation time *T*, and gross reproduction rate *GRR*), and fecundity, were analyzed using TWOSEX-MS Chart software (12/01/2024) [34,35]. Differences between treatments were assessed via the paired bootstrap test [34], which evaluates the significance of differences (*p* < 0.05) based on confidence intervals generated from 100,000 paired comparisons.

The primary formulas are as follows:(1)$$\mathrm{Age}-\mathrm{Specific}\;\mathrm{Survivability}\;(l_{x}):\;l_{x}=\sum_{j=1}^{m}S_{xj}$$(2)$$\mathrm{Age}-\mathrm{Stage}-\mathrm{Specific}\;\mathrm{Fecundity}\;(m_{x}):\;m_{x}=\frac{\sum_{j=1}^{m}S_{xj}f_{xj}}{\sum_{j=1}^{m}S_{xj}}$$(3)$$\mathrm{Age}-\mathrm{Stage}\;\mathrm{Life}\;\mathrm{Expectancy}\;(e_{\mathrm{xj}}):\;e_{xj}=\sum_{i=x}^{\infty }\sum_{y=j}^{m}{S\prime }_{iy}$$(4)$$\mathrm{Intrinsic}\;\mathrm{Rate}\;\mathrm{of}\;\mathrm{Increase}\;(r):\;\sum_{x=0}^{\infty }e^{-r(x+1)}l_{x}m_{x}=1$$(5)$$\mathrm{Finite}\;\mathrm{Rate}\;\mathrm{of}\;\mathrm{Increase}\;(\lambda ):\;{\lambda =e}^{r}$$


(6)
$$\mathrm{Net}\;\mathrm{Reproductive}\;\mathrm{Rate}\;(R_{0}):\;R_{0}=\sum_{x=0}^{\infty }l_{x}m_{x}$$



(7)
$$\mathrm{Mean}\;\mathrm{Generation}\;\mathrm{Time}\;(T):\;T=\frac{\ln R_{0}}{r}$$



(8)
$$\mathrm{Gross}\;\mathrm{reproduction}\;\mathrm{rate}\;(\mathrm{GRR}):\;GRR=\sum m_{x}$$


GraphPad Prism10.3.0 software was adopted for plotting.

## 3. Results

### 3.1. Developmental Period, Longevity, and Fecundity of Leptinotarsa decemlineata in Response to Different Potencies

Significant differences in developmental periods were observed across treatments (*p* < 0.05). Specifically, the developmental periods of first-instar larvae and pupae were prolonged as the cadmium concentration increased. In the control group (0 mg/kg), the developmental periods of the second- and third-instar larvae were longer than those in the cadmium-treated groups. The control group exhibited the highest fecundity (74 days) and longest lifespan (53.76 days). In contrast, fecundity significantly declined across all cadmium treatments as the cadmium concentration increased. Additionally, the pre-oviposition period was apparently extended by higher cadmium concentrations (*p* < 0.001) (Table 1).

### 3.2. Population Parameters of Leptinotarsa decemlineata in Response to Different Potencies

As the cadmium concentration elevated, remarkable reductions were observed in the innate growth rate (*r*), finite growth rate (*λ*), net reproductive rate (*R*_0_), and gross reproduction rate (*GRR*) (*p* < 0.05), while no significant difference was found in the mean generation time (*T*) (*p* > 0.05). Specifically, under B30, C60, D90, and E120 treatments, the innate growth rates were 0.2173 d^−1^, 0.214 d^−1^, 0.018 d^−1^, and 0.0026 d^−1^, respectively; the finite growth rates were 1.021 d^−1^, 1.0216 d^−1^, 0.981 d^−1^, and 0.793 d^−1^, respectively; the net reproductive rates were 2.93, 2.76, 0.41, and 0.28, respectively; and the gross reproduction rates were 11.369, 7.521, 2.03, and 1.812, respectively. These values were all lower than those observed in the control group, which exhibited an innate growth rate of 0.618 d^−1^, finite growth rate of 1.063 d^−1^, net reproductive rate of 16.74, and gross reproduction rate of 21.773 (Table 2).

### 3.3. Age-Stage-Specific Survival Rate

The age-stage-specific survival rate (*S_xi_*) represents the probability of an individual surviving from the egg stage to age *x* at stage *j*. The analysis revealed overlapping survival curves across developmental stages. When the E120 treatment was applied, the probability of *L. decemlineata* developing from the first instar to the adult stage was the lowest, at 16.67%. In response to CK, B30, C60, D90, and E120 treatments, female survival rates (61.6%, 45.8%, 22.5%, 32.5%, and 20%) were all higher than male survival rates (19.16%, 2.5%, 8.3%, 2.5%, and 3.3%). As the cadmium concentration increased, the pupal survival rate gradually declined across all treatments: CK (75%), B30 (72.5%), C60 (63.3%), D90 (59.1%), and E120 (50.0%) (Figure 1).

### 3.4. Age-Specific Survivability and Age-Stage-Specific Fecundity

As depicted in the figure, the survival curves (*l_x_*) of *L. decemlineata* reared under B30, C60, D90, and E120 treatments were markedly lower than those of the control group (CK). The peak values of age-specific fecundity for beetles reared under B30, C60, and E120 conditions (m_56_ = 0.93 eggs; m_50_ = 0.6 eggs; m_28_ = 0.714 eggs) were all below the peak observed under CK (1.194 eggs). Similarly, the highest age-specific reproductive values (*l_x_m_x_*) for beetles reared under B30, C60, D90, and E120 treatments (l_56_m_56_ = 0.315 eggs·d^−1^; l_51_m_51_ = 0.08 eggs·d^−1^; l_44_m_44_ = 0.44 eggs·d^−1^; l_56_m_56_ = 0.127 eggs·d^−1^) were all significantly lower than those observed in the control group (CK: l_51_m_51_ = 0.96 eggs·d^−1^). The peak values of age-specific fecundity for female beetles reared under B30, C60, and E120 conditions (f_566_ = 1.004 eggs·d^−1^; f_506_ = 0.8 eggs·d^−1^; f_566_ = 0.852 eggs·d^−1^) were also lower than those under CK (f_646_ = 1.8 eggs·d^−1^) and D90 (f_446_ = 1.962 eggs·d^−1^) treatments (Figure 2).

### 3.5. Age-Stage-Specific Life Expectancy

The specific age-stage life expectancy (*e_xi_*) reflects the expected survival duration of an individual at age *x* and stage *j*. The life expectancy at the point of first oviposition corresponds to the insect’s average lifespan. The expected lifespan of *L. decemlineata* reared under CK conditions (e_176_ = 45.77 days) significantly exceeded that of beetles reared under B30, C60, D90, and E120 treatments (e_196_ = 36.34 days, e_216_ = 28.86 days, e_196_ = 33.46 days, and e_216_ = 37.62 days) (Figure 3).

### 3.6. Age-Stage-Specific Reproductive Value

The specific age-stage reproductive value (*v_xi_*) denotes the contribution of an individual at age *x* and stage *j* to the future reproductive output of the population. As illustrated in the figure, the peak reproductive values for B30, C60, D90, and E120 treatments (v_406_ = 5.39; v_216_ = 2.47; v_406_ = 8.94; v_216_ = 3.30) were consistently lower than the peak value observed in the control group (CK: v_436_ = 12.73) (Figure 4).

### 3.7. Deformity of Leptinotarsa decemlineata in Response to Different Potencies

Cadmium exposure profoundly influenced the number of deformities in *Leptinotarsa decemlineata* (*p* < 0.001). The numbers of normal beetles reared under B30, C60, D90, and E120 conditions were evidently lower compared to the control group (CK) (*F* = 40.592, *df* = 4, *p* ≤ 0.010), while deformities were markedly higher in beetles exposed to cadmium treatments than in the control group (CK) (*F* = 4.929, *df* = 4, *p* ≤ 0.010) (Figure 5).

## 4. Discussion

In the preliminary phase of this study, exposure to low concentrations of Cd^2+^ (30 mg/kg) promoted potato seedling emergence and growth and increased chlorophyll content, thereby enhancing photosynthetic efficiency. In contrast, exposure to higher concentrations of Cd^2+^ (60 mg/kg, 90 mg/kg, and 120 mg/kg) delayed seedling emergence, inhibited plant growth, reduced photosynthetic activity, and ultimately resulted in decreased yield (Appendix A). These findings underscore the detrimental impact of cadmium on crop growth and its potential to cause significant economic losses. As cadmium accumulates in plant tissues, it enters the food chain and leads to biomagnification [21] upon ingestion by herbivorous insects [36]. Ultimately, this process adversely affects the growth, development, and population dynamics of herbivorous insects [9].

Similarly, *Spodoptera litura* larvae exposed to cadmium exhibited delayed development. After feeding on cadmium-contaminated *Populus alba* leaves, their body length and pupal weight were both reduced [24]. This finding aligns with the results of this study, where the developmental periods of *L. decemlineata* larvae (first instar) and pupae, as well as the pre-oviposition period, were extended as the cadmium concentration increased. However, contrary to the findings of *S*. *litura*, the second- and third-instar larvae in the control group (0 mg/kg) had longer developmental periods compared to those in the cadmium-treated groups (Table 1). This discrepancy may be attributed to the preferential feeding behavior of herbivorous insects, as they prefer plants with lower water content [37]. As the cadmium concentration increased, the water content of potato leaves dropped significantly. Given insects’ non-selective feeding behavior on cadmium-contaminated plants [38,39], this reduction in leaf water content likely led to an increased feeding rate among *L. decemlineata* larvae, promoting rapid growth and shortening the developmental periods of the second- and third-instar larvae.

Previous studies have shown that cadmium exposure reduced the hatching rate, weight, and survival rate of *Musca domestica* [25]. In *Drosophila melanogaster*, exposure to cadmium concentrations of 15.0 mg/L, 30.0 mg/L, and 60.0 mg/L resulted in a shortened lifespan and reduced reproductive capacity, and the severity of these effects was positively correlated with the concentration of cadmium [26]. These observations are consistent with the findings of this study, in which the control group exhibited the highest reproductive capacity (74 days) and longest lifespan (53.76 days), and reproductive capacity across all cadmium-treated groups significantly reduced with the cadmium concentration (Table 1, Figure 2, Figure 3 and Figure 4).

Moreover, cadmium exposure can induce teratogenic effects and influence insects’ behavior. Various deformities in insects’ mouthparts have been observed. Cadmium causes mouth malformation [40] in insects, which is consistent with the results in this study, which indicated that the presence of cadmium increased the prevalence of malformation in *L. decemlineata* (Figure 5). It is not difficult to speculate that the malformations caused by cadmium will also affect the growth and development of insects.

Sun et al. found that the concentration of heavy metals in the bodies of *Spodoptera litura* increased across generations, with both larval survival and pupation rates being lower than those of the control group [41]. This observation is consistent with the findings in this study (Figure 1 and Figure 2), suggesting that heavy metal exposure impedes the growth and development of *L. decemlineata*.

It has been demonstrated that *Sitobion avenae* exhibits heightened sensitivity to plants grown in cadmium-contaminated soil (20 mg/kg). The life table parameters of this aphid, including its innate growth rate, finite rate of increase, and net reproductive rate, decline as a result of feeding on contaminated plants, and adult fecundity is also significantly reduced [42]. In this study, increasing cadmium concentrations in soil resulted in lower fecundity, which in turn led to a reduction in the innate growth rate (r), finite rate of increase (λ), net reproduction rate (R_0_), and gross reproduction rate (GRR). These findings align with those of Shu [43], indicating that cadmium has a detrimental effect on the population dynamics of *L. decemlineata*.

In contrast, Song Yaqian investigated the effects of combined cadmium and zinc exposure on *Sitobion avenae* across multiple generations and reported that the net reproductive rate increased in the 1st, 3rd, 5th, and 10th generations. This finding diverges from the results of this study, in which the net reproduction rate of *L. decemlineata* decreased with increasing cadmium concentrations. However, Song Yaqian observed no significant effect on the average generation time, which agrees with this study [27] (Table 2).

## 5. Conclusions

This study demonstrates that cadmium stress significantly prolongs the developmental duration of certain life stages of *L. decemlineata* and reduces its lifespan and fecundity, thereby inhibiting population growth. Moreover, exposure to cadmium results in an increased incidence of deformities in the beetles post-eclosion. Our research provides a preliminary understanding of the transfer and transformation of cadmium from soil to plants and, ultimately, to herbivorous insects, revealing its detrimental effects on the food chain. However, the accumulation of cadmium in adult *L. decemlineata* and its potential impacts on natural enemy populations remain unaddressed and necessitate further investigation.

## Figures and Tables

**Figure 1 insects-16-00073-f001:**
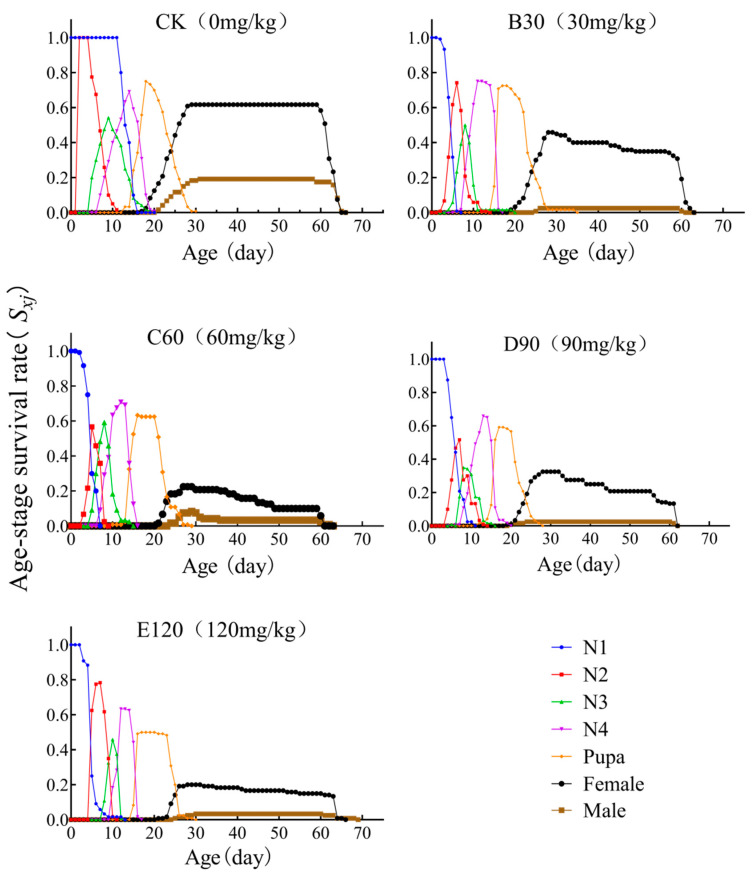
Age-stage-specific survival rate (*S_xj_*) of *Leptinotarsa decemlineata* on heavy metal cadmium-stressed potato plants. Note: N1–N5 represent the first, second, third, fourth, and fifth nymph stages, respectively.

**Figure 2 insects-16-00073-f002:**
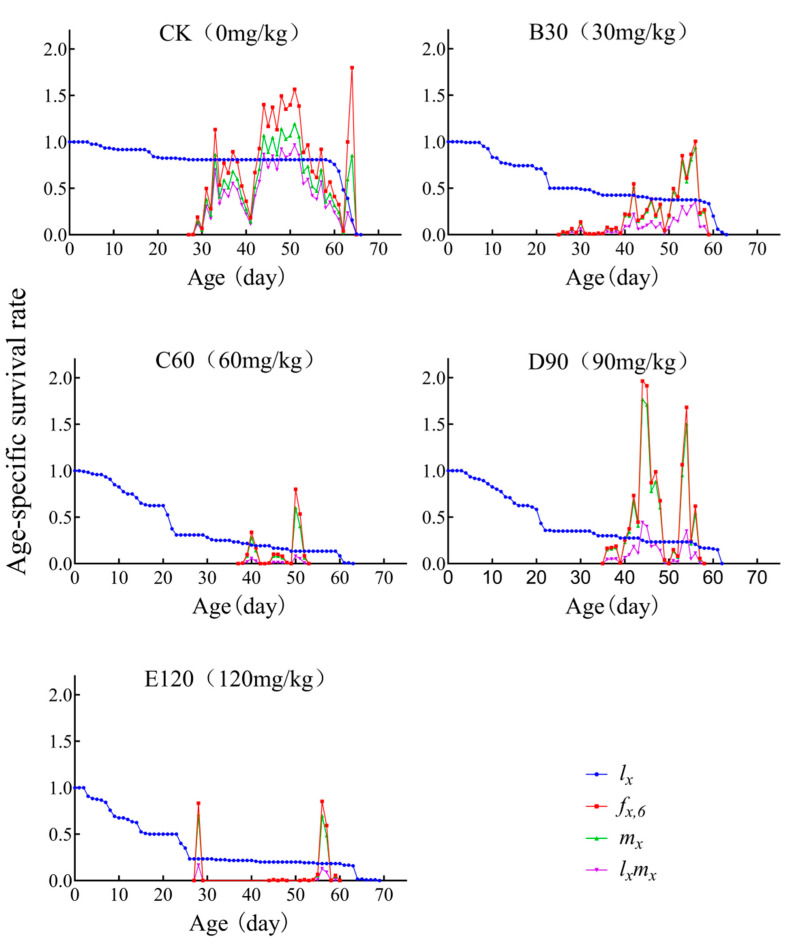
Age-specific survival rate (*l_x_*), female age-specific fecundity (*f_x_,*_6_), age-specific fecundity of total population (*m_x_*), and age-specific maternity (*l_x_m_x_*) of *Leptinotarsa decemlineata* on heavy metal cadmium-stressed potato plants.

**Figure 3 insects-16-00073-f003:**
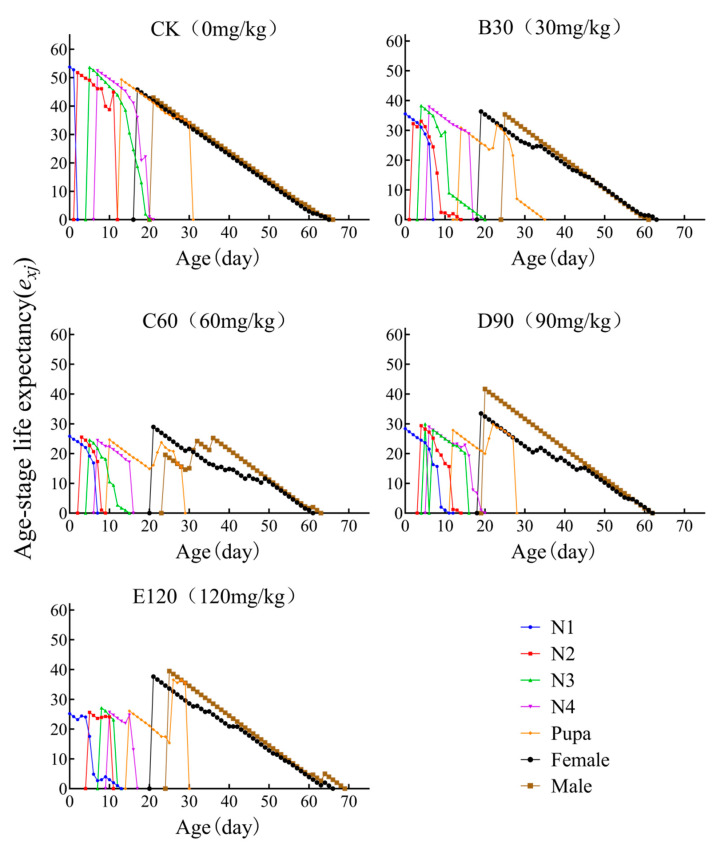
Age-stage life expectancy (*e_xj_*) of *Leptinotarsa decemlineata* on heavy metal cadmium-stressed potato plants. Note: N1–N5 represent the first, second, third, fourth, and fifth nymph stages, respectively.

**Figure 4 insects-16-00073-f004:**
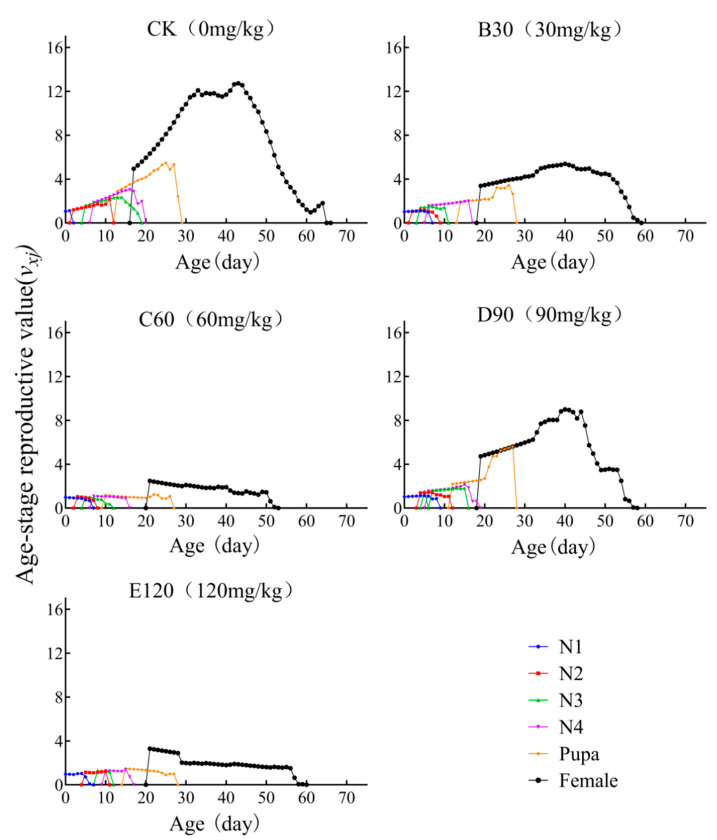
Age-stage reproductive value (*v_xj_*) of *Leptinotarsa decemlineata* on heavy metal cadmium-stressed potato plants. Note: N1–N5 represent the first, second, third, fourth, and fifth nymph stages, respectively.

**Figure 5 insects-16-00073-f005:**
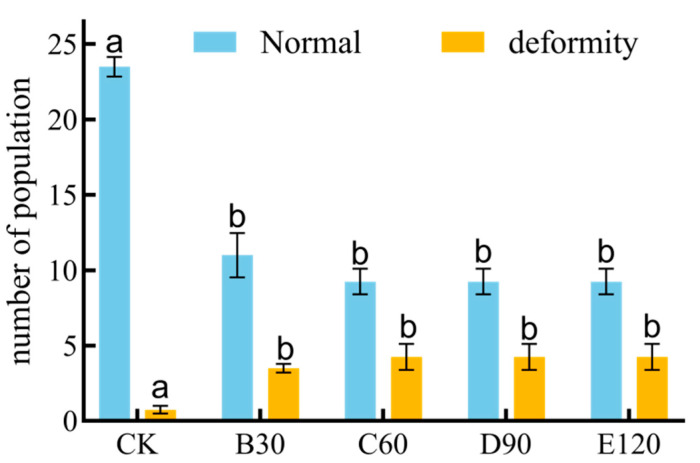
Deformity of *Leptinotarsa decemlineata* on cadmium-stressed potato plants. Note: Data in the figure were average rate ± standard error; Different lowercase letters in the figure indicated significant differences among different Potencies (*p* < 0.05) (one-Way ANOVA).

**Table 1 insects-16-00073-t001:** Biological characteristics of *Leptinotarsa decemlineata* on heavy metal cadmium-stressed potato plants.

Developmental Period at Different Potency					
Developmental Stage	0 mg/kg (CK)	30 mg/kg (B30)	60 mg/kg (C60)	90 mg/kg (D90)	120 mg/kg (E120)
1st Instar	2.00 ± 0.00 a	4.91 ± 0.09 b	5.19 ± 0.11 c	6.30 ± 0.15 d	6.51 ± 0.04 e
2nd Instar	5.34 ± 0.17 a	2.41 ± 0.13 c	1.71 ± 0.06 d	2.43 ± 0.13 c	4.19 ± 0.08 b
3rd Instar	4.20 ± 0.14 a	2.07 ± 0.05 c	2.25 ± 0.09 b	2.11 ± 0.12 bc	1.95 ± 0.11 d
4th Instar	5.54 ± 0.19 b	6.54 ± 0.11 a	5.66 ± 0.14 b	5.79 ± 0.20 c	5.73 ± 0.10 d
Pupal stage	7.34 ± 0.23 a	8.79 ± 0.28 c	9.68 ± 0.32 e	8.26 ± 0.33 b	9.46 ± 0.27 d
Adult stage	39.18 ± 0.21 a	31.12 ± 1.44 c	24.00 ± 2.13 e	29.36 ± 1.70 d	34.36 ± 1.82 b
TPOP	31.83 ± 0.17 a	36.43 ± 0.78 b	39.75 ± 0.65 d	39.45 ± 0.63 c	53.52 ± 1.46 e
Fecundity	74.00 ± 1.2 a	55.00 ± 0.72 b	27.00 ± 0.43 c	22.00 ± 1.63 d	21.00 ± 1.01 e
Total longevity	53.76 ± 1.77 a	35.54 ± 1.93 b	25.82 ± 1.57 d	28.37 ± 1.81 c	25.18 ± 1.89 e

Note: Significant difference was analyzed by paired-sample paired bootstrap test of TWOSEX—MS Chart. Data were mean ± SE. Different letters in the same row indicated significant difference at *p* < 0.05 level by paired bootstrap test.

**Table 2 insects-16-00073-t002:** Population parameters of *Leptinotarsa decemlineata* on heavy metal cadmium-stressed potato plants.

Population Parameters of Different Potencies					
Population Parameters	0 mg/kg (CK)	30 mg/kg (B30)	60 mg/kg (C60)	90 mg/kg (D90)	120 mg/kg (E120)
Intrinsic rate of increase (*r*) (d^−1^)	0.618 ± 0.130 a	0.2173 ± 0.304 b	0.214 ± 0.00 b	0.018 ± 0.00 c	0.0026 ± 0.001 c
Finite rate of increase (*λ*) (d^−1^)	1.063 ± 2.265 a	1.021 ± 3.067 b	1.0216 ± 4.855 b	0.981 ± 8.579 c	0.793 ± 9.139 c
Net reproductive rate (*R*_0_)	16.74 ± 1.41 a	2.93 ± 0.44 b	2.76 ± 0.61 b	0.41 ± 0.19 c	0.28 ± 0.11 c
Gross reproduction rate (*GRR*)	21.773 ± 1.59 a	11.369 ± 1.82 b	7.521 ± 0.79 c	2.030 ± 0.85 d	1.812 ± 0.61 e
Mean generation time (*T*) (d)	49.427 ± 0.502 a	47.477 ± 7.073 a	47.549 ± 0.729 a	47.406 ± 0.729 a	45.545 ± 0.368 a

Note: Significant difference was analyzed by a paired-sample paired bootstrap test of TWOSEX—MS Chart. Data were mean ± SE. Different letters in the same row indicated significant difference at *p* < 0.05 level by paired bootstrap test.

## Data Availability

Data will be made available on request.

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
