# Peer review of "Age-Stage, Two-Sex Life Table of Leptinotarsa decemlineata (Coleoptera: Chrysomelidae) Experiencing Cadmium Stress"

_insects, 2025, doi:10.3390/insects16010073_

Round 1

Reviewer 1 Report

Comments and Suggestions for Authors

The manuscript, entitled "Age-Stage, Two-Sex Life Table of Leptinotarsa decemlineata (Coleoptera: Chrysomelidae) at Cadmium Stress" is an intriguing idea, and I appreciate the effort and thought that went into your study. Below are some questions and detailed suggestions and corrections aimed at improving the clarity, coherence, and scientific rigor of your manuscript:

  • The idea of assessing the fitness of Leptinotarsa decemlineata under cadmium stress using a two-sex life table approach is indeed novel. However, please provide a clear justification for conducting this study.

  • Highlight the benefits of using the two-sex life table method to analyze the fitness impact of cadmium exposure on this pest species.

  • Emphasize the significance of cadmium as a major contaminant in agricultural soils, which affects economic plants, including potatoes. This will establish the broader relevance of your research.

To improve the sequence and flow of your introduction, consider restructuring it as follows:

  1. Begin with the global importance of potatoes as a staple crop, particularly focusing on their significance in China.

  2. Introduce the economic status and biological importance of the Colorado potato beetle (Leptinotarsa decemlineata), mentioning its classification under Coleoptera: Chrysomelidae.

  3. Discuss the role of cadmium contamination in agriculture, including its impact on soil health and crop productivity.

  4. Finally, clearly state your objectives for conducting this study.

    • Provide the family and order of L. decemlineata early in the introduction.

    • Include damage percentage values caused by this pest in China and worldwide, if available.

    • Avoid bold text; instead, ensure the objectives are presented concisely and clearly at the end of the introduction.

  • When conducting laboratory experiments, ensure you use a pure lab culture reared on a specific diet. Explain the rationale for conducting experiments on the second generation of the population.

  • Justify how you ensured the purity of your insect population. This is crucial for the credibility of your experimental setup.

  • Provide a clear explanation for choosing the specific cadmium doses and discuss their ecological relevance.

  • Address how long the insects were exposed to the cadmium-treated diet to effectively assess its impact.

  • Detail the method for differentiating between larval instars and the process for counting eggs. Clarify if pair replication was conducted, and explain how data on eggs were collected.

Discussion Section

  • The discussion should be structured step by step, avoiding any repetition of the methodology. Link your findings to previous research to strengthen your arguments.

  • Clearly explain the implications of your results and how they contribute to understanding the impact of cadmium stress on pest populations.

  • Carefully check the symbols used in your manuscript against the standards outlined in the latest publications by Hsin Chi. Ensure all symbols align with the guidelines.

  • Read and cite recent papers by Hsin Chi to validate and support your methodology and findings.

  • Avoid repeating the methodology in the discussion section.

  • Ensure all figures, tables, and symbols are accurate and consistent throughout the manuscript.

  • Include any relevant damage percentage values caused by L. decemlineata in both China and globally.

I commend you on the potential of this study to contribute significantly to understanding the ecological impact of heavy metal stress on pest fitness. With these revisions, your manuscript will be better positioned for publication.

Please check the annotated pdf file.

Comments on the Quality of English Language

Please consult a native English speaker or a technical expert to improve the language and clarity of your manuscript.

Reviewer 2 Report

Comments and Suggestions for Authors

The MS entitled with Age-Stage, Two-Sex Life Table of Leptinotarsa decemlineata (Coleoptera: Chrysomelidae) at cadmium stress investigated the life parameters of Colorado potato beetle under cadmium stress. It will provide some evidence for the pest control and food security. It was well presentation, and the data was solid. However, it should be revised before its publication. The comments are as follows:

1.     The first concern is that the significance for this study is not clear. Now that cadmium is a threat to crop health, what will be suggested in further pest control of Colorado potato beetle according to this study? Is it that we could use cadmium to control the beetles?

2.     Meanwhile, cadmium will also affect the growth of potato plant, what is growth phenotype for the potatoes under cadmium stress in this study? The detrimental effects observed for the beetles could also attribute to the poor feeding potato leaves. How to determine whether these effects are directly from cadmium or from plants stressed by cadmium?

3.     Does it appropriate to analyze these parameters with one-way ANOVA method? For that these data did not follow a normal distribution.

4.     Why choose these the four cadmium concentrations in this study?

5.     How to choose a positive control in this study?

Round 2

Reviewer 1 Report

Comments and Suggestions for Authors

Well but, please remove your manuscript plagerism.

Reviewer 2 Report

Comments and Suggestions for Authors

I have no more concerns about this version.